# Whole blood transcriptomic analysis of beef cattle at arrival identifies potential predictive molecules and mechanisms that indicate animals that naturally resist bovine respiratory disease

**Matthew A. Scott** [1]*, **Amelia R. Woolums**[1], **Cyprianna E. Swiderski**[2], **Andy D. Perkins** [3], **Bindu Nanduri**[4], **David R. Smith**[1], **Brandi B. Karisch**[5], **William B. Epperson**[1], **John R. Blanton Jr.**[5]

**1** Department of Pathobiology and Population Medicine, Mississippi State University, Mississippi State, MS, United States of America, **2** Department of Clinical Sciences, Mississippi State University, Mississippi State, MS, United States of America, **3** Department of Computer Science and Engineering, Mississippi State University, Mississippi State, MS, United States of America, **4** Department of Basic Sciences, Mississippi State University College of Veterinary Medicine, Mississippi State University, Mississippi State, MS, United States of America, **5** Department of Animal and Dairy Sciences, Mississippi State University, Mississippi State, MS, United States of America

\* mas1052@msstate.edu

**Data Availability Statement:** The sequence data underlying the results presented in this study were

## Abstract

Bovine respiratory disease (BRD) is a multifactorial disease complex and the leading infectious disease in post-weaned beef cattle. Clinical manifestations of BRD are recognized in beef calves within a high-risk setting, commonly associated with weaning, shipping, and novel feeding and housing environments. However, the understanding of complex host immune interactions and genomic mechanisms involved in BRD susceptibility remain elusive. Utilizing high-throughput RNA-sequencing, we contrasted the at-arrival blood transcriptomes of 6 beef cattle that ultimately developed BRD against 5 beef cattle that remained healthy within the same herd, differentiating BRD diagnosis from production metadata and treatment records. We identified 135 differentially expressed genes (DEGs) using the differential gene expression tools edgeR and DESeq2. Thirty-six of the DEGs shared between these two analysis platforms were prioritized for investigation of their relevance to infectious disease resistance using WebGestalt, STRING, and Reactome. Biological processes related to inflammatory response, immunological defense, lipoxin metabolism, and macrophage function were identified. Production of specialized pro-resolvin mediators (SPMs) and endogenous metabolism of angiotensinogen were increased in animals that resisted BRD. Protein-protein interaction modeling of gene products with significantly higher expression in cattle that naturally acquire BRD identified molecular processes involving microbial killing. Accordingly, identification of DEGs in whole blood at arrival revealed a clear distinction between calves that went on to develop BRD and those that resisted BRD. These results provide novel insight into host immune factors that are present at the time of arrival that confer protection from BRD.

submitted to the National Center for Biotechnology Information Gene Expression Omnibus under GSE136176.

**Funding:** This project was supported by the Mississippi State University Department of Pathobiology and Population Medicine, Mississippi State University Department of Animal and Dairy Sciences, and the Mississippi Agricultural and Forestry Experiment Station. The funders had no role in study design, data collection and analysis, decision to publish, or preparation of the manuscript.

**Competing interests:** The authors have declared that no competing interests exist.

## Introduction

Bovine respiratory disease (BRD) is the leading cause of morbidity and mortality in cattle in the United States [1]. BRD is a multifactorial disease complex; causative factors include interactions between infectious etiological agents, host immune response, and environmental risk factors [2] [3] [4]. The syndrome is often recognized in young, newly weaned beef cattle experiencing the stresses of weaning, shipping, and novel feeding and/or housing environments [5] [6] [7]. The infectious etiological agents associated with BRD and disease-mitigating management protocols have been studied extensively; however, understanding of the complex interactions between these factors and how the genomic background of individual animals influences disease susceptibility remains elusive [8] [9] [10] [11] [12] [13]. Additionally, diagnosis of BRD remains imprecise and is often established based on clinical signs including elevated rectal temperature, depression, anorexia, and nasal discharge [14] [15] [16]. The investigation of genetic and molecular mechanisms related to BRD susceptibility or resistance could lead to discovery of biomarkers that enable more accurate BRD diagnosis.

Because of the dynamic complexity of immunological defense against BRD and the wide assortment of factors influencing morbidity, emerging technologies have promise for advancing disease identification and prediction. For example, genome-wide association studies (GWAS) have been implemented to identify quantitative trait loci (QTLs) associated with resistance to BRD. Candidate genomic areas related to cell adhesion activity, fibrinolysis, and inflammatory mediation have been identified [17] [18] [19]. Although studies involving SNP genotyping of cattle offer encouraging observations of trait-loci relationships, GWAS alone have limitations. SNP prediction can be over-estimated due to sample size bias and breed genomic similarity, which often cannot be corrected with independent validation testing [20] [21]. Additionally, SNP association without gene expression data often cannot account for joint interaction of multiple genes or gene-protein interactions [22] [23].

There is evidence that the dynamic interactions that characterize disease and host immune factors can be elucidated using transcriptomic analysis via RNA-Seq, in order to identify relationships between host gene expression and BRD outcome. As a complement to GWAS, RNA-Seq analysis provides a highly sensitive methodology for transcript expression detection, without need for prior knowledge of the genome [24]. Recently, whole blood RNA-Seq has been used to discover gene expression signatures for clinical phenotyping of patients affected by human rhinovirus and acute respiratory distress syndrome (ARDS) [25] [26]. These studies demonstrate that whole blood RNA-Seq analysis can be used to identify biomarkers that predict or diagnose respiratory disease. However, to our knowledge, there has been no study that has identified differentially expressed gene products in whole blood of cattle at high risk for BRD.

This study profiles the whole blood transcriptome in post-weaned beef cattle that went on to develop BRD and also in cattle that ultimately resisted BRD. These blood samples were collected when the cattle were first purchased and before disease was identified. Our aim in the timing of this blood collection was to identify gene products that characterize the biological status of typical calves which have been recently weaned, transported, and co-mingled prior to facility arrival in order to identify gene products associated with BRD resistance. By comparing DEGs in cattle that developed BRD versus those that did not, we provide a characterization of biological interactions that may contribute to both the development of and resistance to BRD.

## Materials and methods

### Animal use and management

This research was approved by the Mississippi State University Institutional Animal Care and Use Committee (IACUC protocol #17–120). This experiment was a subset of a larger study

focused on examining the effect of on-arrival vaccination and deworming on health and performance outcomes [27]. Eighty crossbred steers (n = 16) and bulls (n = 64) were received from local livestock auctions over a two-day period (day -3, day -2). On day 0, bull calves were surgically castrated, and all animals were given identification ear tags. All cattle were tested for persistent infection with bovine viral diarrhea virus (BVDV) via ear notch antigen capture ELISA; no BVDV-positive calves were identified. Animals were randomly placed into twenty pens to assure even distribution of body weight and fecal egg counts. Calves were weighed on days 0, 14, 28, 42, 56, 70, and 84, and average daily gain was calculated every 14 days and for the entire study.

Animals were managed daily by staff trained to identify medical problems including lameness and signs of BRD. Clinical signs of BRD were assigned severity scores of 1–4 based on visual inspection using an adapted approach described by Step et al. [28]. Animals given a score of 1 or 2, which also had a rectal temperature $\geq 40°$ C, or a score of 3 or 4, regardless of rectal temperature, were treated with antimicrobials. The use of antimicrobial therapy in this study has been described by Woolums et al. [29] Briefly, first-time treatment for any calf that was clinically diagnosed with BRD was ceftiofur crystalline free acid (Excede, Zoetis) at 1.5 mg/kg subcutaneously (SC) at the base of the ear, given once. Animals diagnosed with BRD a second time, 7 days or greater after the first treatment, were treated with florfenicol (Nuflor, Merck Animal Health) at 40 mg/kg SC, given once. Animals diagnosed with BRD a third time, 4 days or greater after the second treatment, were given oxytetracycline (Noromycin 300 LA, Norbrook) at 20 mg/kg SC, given once. Cattle whose signs of BRD persisted following treatment with oxytetracycline were monitored carefully for pre-determined endpoints indicating that the animal was unlikely to recover followed by euthanasia if deemed appropriate by project veterinarians.

### Initial animal selection

Forty of the 80 cattle enrolled in the larger study were not dewormed at arrival (day 0). 24 of these 40 non-dewormed cattle were randomly selected via the RANDBETWEEN function in Excel (Microsoft) for at-arrival (day 0) whole blood sampling via the jugular vein. Blood was collected into Tempus tubes (Applied Biosystems). As this investigation was a subset of a larger study designed to examine vaccination and deworming effects on health and performance [27], 12 of the 24 cattle that had blood collected were not vaccinated at arrival. The remaining 12 cattle that had blood collected were vaccinated (day 0) using modified live virus vaccines against bovine herpesvirus-1, bovine viral diarrhea virus types 1 and 2, parainfluenza type 3 virus, and bovine respiratory syncytial virus (Express 5, Boehringer Ingelheim Vetmedica). Those 12 cattle also received vaccination against *Clostridium chauvoei*, *septicum*, *novyi*, *sordelli*, and *perfringens* types C and D (Vision 6 with SPUR, Merck Animal Health). Vaccines were given SC. Blood tubes were stored at -80°C until analysis.

Production and treatment records of all animals were recorded throughout the course of the study. After the study was completed, 11 of the 24 randomly sampled animals were identified to have been treated for BRD within the first 28 days of the study. The remaining 13 of the 24 randomly sampled animals were never identified to have signs of BRD; for simplicity these cattle are subsequently referred to as "healthy". Six healthy and 6 BRD animals from the 24 randomly sampled cattle were selected for RNA sequencing, based on even distribution of vaccination status at arrival between the two groups; 3 animals in each group had been vaccinated after blood collection on day 0 and 3 had not. None of the healthy cattle died, while 3 of the 6 cattle in the BRD group died of their naturally occurring BRD in spite of treatment; two died on study day 17 and one died on study day 51. Necropsy of the three cattle at the time of their

death confirmed the diagnosis of BRD, with all 3 animals testing positive for *Mannheimia hae-molytica* upon bacterial lung culture. The average daily weight gain (ADG) for the 84-day trial was higher in healthy cattle than BRD cattle. More information about the cattle is presented in S1 Table.

## RNA extraction and sequencing

RNA extraction, concentration and quality evaluation, along with library preparation, and RNA sequencing were performed by the UCLA Technology Center for Genomics and Bioinformatics (UCLA TCGB, Los Angeles, CA, USA). mRNA purification was performed using the Tempus Spin RNA Isolation Kit (Applied Biosystems). mRNA quality and concentrations were measured using Agilent 2100 Bioanalyzer (Agilent). All mRNA samples were of high quality (RIN: 7.4–9.7, mean = 9.0). Paired-end cDNA libraries were generated (TruSeq stranded mRNA, Illumina) and sequencing was performed using an Illumina HiSeq 3000 (Illumina, v3.3.76; SBS reagent kit) in 2 × 150 base pair length reads in two lanes, at 80M reads per sample.

## Data processing and RNA-Seq analysis

Raw sequencing reads were pre-processed using FastQC software v0.11.8 to assess read quality [30]. Reads were quality filtered and trimmed using Trimmomatic v0.38 [31]. Leading and trailing bases of each read were removed if they were below a base quality score of 3. Trimming was performed by scanning each read with a 4-base pair sliding window and removing read segments below a minimum base quality score of 15. Finally, only sequences with a minimum read length of 36 bases were kept for read mapping. Trimmed reads were processed and mapped to the bovine reference genome assembly ARS-UCD1.2 [32] using HISAT2 v2.1.0 [33]. Trimmed read and mapping alignment statistics are provided in in S2 Table. An index assembly was created using the hisat2-build function, allowing for the alignment of reads to the bovine reference genome assembly. Mapped reads in sequence alignment/map format (.sam) were converted to binary alignment/map format (.bam) with SAMtools [34] [35]. Transcript/gene assembly and quantification were performed using StringTie v1.3.4 [36] [37]. Assembly tracking and evaluation were classified using GffCompare v0.11.2 [38]. After assembly, a gene-level count matrix was generated from each sample using Python v2.7.16, utilizing the program prepDE.py [39]. One sample (S_72), from the healthy group, was removed from further analysis due to low read count quantity.

Differential gene expression analysis was conducted in R using two tools from the Bioconductor R-package: edgeR v3.24.3 [40] [41] and DESeq2 v1.22.2 [42]. Animals were grouped and factored based on BRD status, and each replicate was placed into a "BRD" or "Healthy" category. Pre-processing of gene counts in edgeR was performed using the filterByExpr package in edgeR, with default settings, in order to retain genes which have an adequate count for statistical assessment. Low read counts in DESeq2 were processed by removing genes with a sum of less than 10 counts across all samples. Gene products identified as differentially expressed with both edgeR and DESeq2 were used for downstream analysis. Both programs use a negative binomial distribution of the read count data in comparing groups, but differ in normalization methodology [43] [44] [45]. Multidimensional scaling was applied to the gene expression data after count filtering, using the plotMDS function from the edgeR package (Fig 2). Identification of DEGs was performed using likelihood ratio testing with a false discovery rate (FDR) of ≤ 0.10 [46]. Heatmapping was performed with the R package pheatmap v1.0.12 [47].

## Overrepresentation analysis

Gene ontology (GO) and biological pathway overrepresentation analysis of the DEGs were both performed in the WebGestalt 2019 (WEB-based GEne SeT AnaLysis Toolkit) online database, using human orthologs of the 36 bovine gene products[48] [49]. After removal of 8 pseudogenes and uncharacterized genes, 28 DEGs were analyzed to identify overrepresented GO biological pathways. Biological pathway overrepresentation was also performed using these 28 DEGs in WebGestalt 2019, utilizing the Reactome database [50]. All analyses with an adjusted p-value (FDR) of $\leq 0.10$ were considered significantly enriched.

## Network analysis of DEGs

Protein-protein interactions were evaluated using Search Tool for the Retrieval of Interacting Genes (STRING) database v11.0, by employing the same human orthologs of the DEGs used in the overrepresentation analysis [51]. Networks of functional partners were constructed to identify molecular interactions and regulatory gene products. STRING uses text mining, experimental data, database searches, co-expression, and physical interaction information as sources for interaction criteria. Interactions were considered relevant if they had medium or greater confidence as defined in STRING. Protein-protein interaction modeling was performed in two parts: 1) to determine if gene products higher in expression in healthy cattle formed functional networks and 2) to determine if gene products higher in expression in BRD animals formed functional networks with the addition of predicted interactors. Network clustering of the DEGs higher in expression in healthy animals was performed with the k-means algorithm within the STRING interface, empirically pre-set to 4 clusters, with omission of disconnected nodes (i.e. gene products).

## RT-qPCR validation

Four gene products identified as differentially expressed by RNA-seq were selected for validation in the study cohort of healthy and diseased cattle. Expression of IL5RA, GATA2, ALOX15, and HPGD were quantified using real-time quantitative polymerase chain reaction (RT-qPCR). DDX31 and UBE2Q1 were employed as housekeeping genes in RT-qPCR reactions. These housekeeping genes were selected using NormFinder, based upon consistent expression across diseased and non-diseased cattle in the previously detailed RNA-Seq dataset [52]. Total RNA was isolated from stored whole blood samples from the validation cohort using the Tempus Spin RNA Isolation Kit protocol (ThermoFisher Scientific). RNA concentration was assessed with a NanoDrop 8000 Spectrophotometer (ThermoFisher Scientific); 1000ng of RNA was reversed transcribed into cDNA with qScript cDNA SuperMix, using the kit protocol (Quanta Biosciences). Quantitative PCR was performed with PerfeCTa SYBR Green FastMix, Low ROX (Quanta Biosciences) in an Applied Biosystems 7500 Fast Real-Time PCR System (ThermoFisher Scientific), using a 40-cycle, two-step protocol with 50 ng of RNA in each reaction well. All reactions were performed in triplicate. Melting curve analysis was performed to validate the specificity of all amplifications. Relative gene expression was analyzed with the $2^{-\Delta\Delta Ct}$ method [53]. Primers were developed with the online tool Primer-BLAST and sequences can be found in S3 Table [54]. $Log_2$ fold changes were calculated for BRD animals relative to the healthy animals. Pearson correlation coefficient was calculated between the fold changes identified in RNA-Seq and RT-qPCR analysis. Student's t-testing was performed for determining significance in RT-qPCR fold changes between healthy and BRD groups. Differences were considered statistically significant with a p-value $\leq 0.05$.

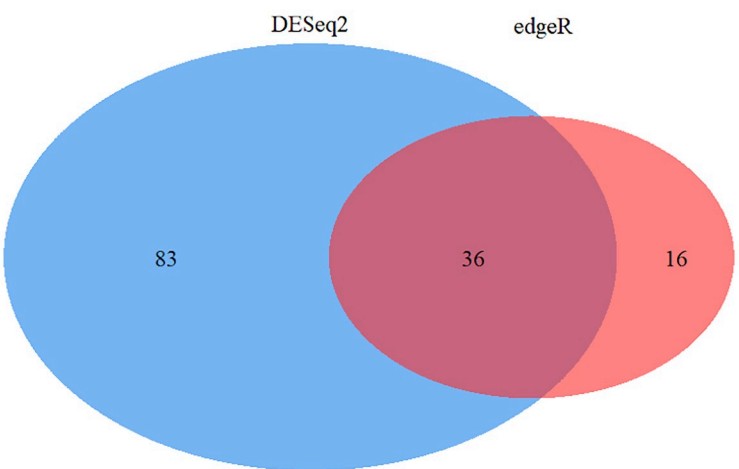

**Fig 1. Number of DEGs identified in high risk beef cattle at arrival.** Venn diagram of the number of differentially expressed genes found in edgeR and DESeq2 analysis.

## Results

### Identification of DEGs in cattle at arrival

A total of 135 gene products were differentially expressed between healthy and BRD animals, 36 of which were identified as differentially expressed with both edgeR and DESeq2 (Fig 1). Complete lists of DEGs from edgeR and DESeq2 analysis, and their intersection, are provided in S4 Table. To visualize the level of similarity between individual cattle based on their relative gene expression patterns, multidimensional scaling (MDS) was applied to gene expression data from the 6 BRD and 5 healthy individuals (Fig 2). Each point on x- and y- axes represents each animal and their transformed Euclidean distance in two dimensions, discerned as leading log2-fold change between the genes that best differentiate each animal. Clustering of gene expression in healthy samples is evident, while gene expression patterns in BRD animals are clearly more dissimilar.

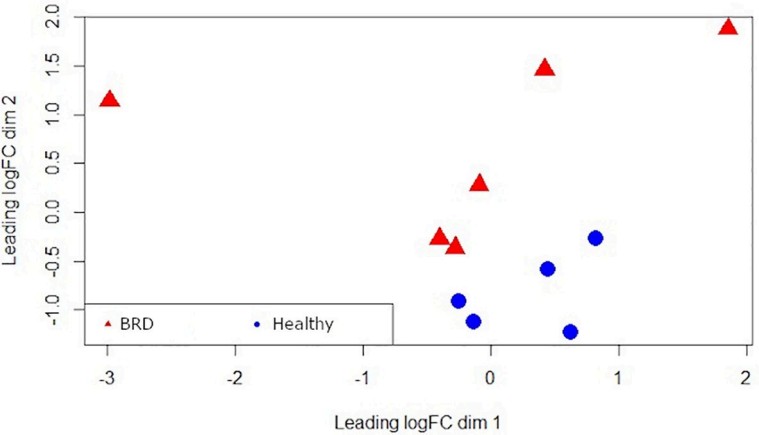

**Fig 2. Multi-dimensional scaling (MDS) plot of the gene count data set from all 11 samples.** Red triangles indicate BRD cattle. Blue circles indicate healthy cattle.

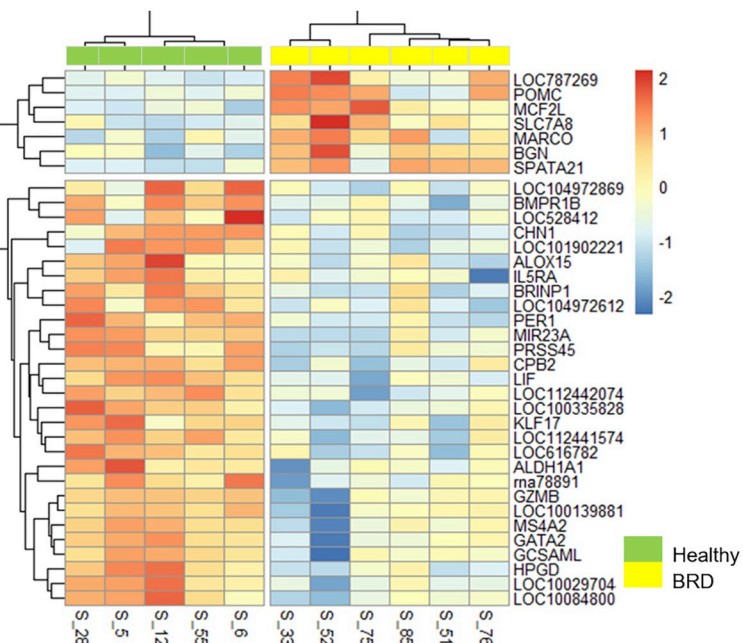

**Fig 3. Heatmap of 36 differentially expressed genes identified at arrival between beef cattle that remained healthy and cattle that later developed BRD.** Color scale (red-to-blue) represents gene expression levels per sample; red and blue colors indicate increased expression and decreased expression, respectively. Note that gene hierarchical clustering of gene expression profiles segregates BRD (yellow top panel) from healthy cattle (green top panel).

A heat map of the 36 DEGs that were common to both edgeR and DESeq2 was generated to visualize expression patterns across all 11 individuals (6 BRD and 5 healthy) (Fig 3). Expression patterns in this heat map are accompanied by hierarchical clustering of genes (rows) and samples (columns), where gene-wise variation was standardized using z-score statistics from trimmed mean of M-values (TMM) normalization. Hierarchical clustering of samples based on expression patterns of DEGs grouped BRD cattle into one cluster and healthy animals into a separate cluster, substantiating a clear distinction between healthy and BRD cattle based on gene expression for these 36 gene products.

## Overrepresentation analysis of DEGs

GO term enrichment analysis performed in WebGestalt 2019 identified over-represented biological processes from the DEGs that were common to edgeR and DESeq2 analyses. We identified 7 significantly overrepresented biological processes, using an FDR-adjusted p-value $\leq 0.10$ (Table 1). The significant biological processes were inflammatory response, defense response, regulation of macrophage differentiation, lipoxin metabolic process, lipoxin biosynthetic process, regulation of inflammatory response, and macrophage differentiation. Primarily, these processes are related to cellular inflammatory responses and leukocyte immunophysiology. These biological processes were predominantly enriched for genes that were higher in expression in healthy animals.

Using WebGestalt 2019 with the selection of the functional database Reactome, biological signaling pathways that were over-represented by the DEGs common to edgeR and DESeq2 analysis were identified. Five pathways were identified as significantly enriched (FDR $\leq 0.10$; Table 2). The significant pathways were the biosynthesis of E-series 18(S)-resolvins, biosynthesis of eicosapentaenoic acid (EPA)-derived specialized pro-resolvin mediators (SPMs),

**Table 1. Significantly enriched biological processes (GO-BP) with associated DEGs in cattle at arrival.**

| Gene Set | Description | Size | Expect | Ratio | P-Value | FDR | Genes |
|---|---|---|---|---|---|---|---|
| GO:0006954 | Inflammatory response | 717 | 1.1616 | 9.4698 | 6.01E-09 | 0.0001 | ALOX15, BMPR1B, CCL14, CD200R1, CPB2, HIST1H2BA, IL5RA, LOC100139881, MS4A2, PER1 |
| GO:0006952 | Defense response | 1518 | 2.4593 | 5.6928 | 1.64E-08 | 0.0001 | ALOX15, BMPR1B, BRINP1, CCL14, CD200R1, CPB2, GZMB, HIST1H2BA, IL5RA, LOC100139881 |
| GO:0045649 | Regulation of macrophage differentiation | 21 | 0.034 | 88.18 | 4.95E-06 | 0.0196 | GATA2, LIF, RB1 |
| GO:2001300 | Lipoxin metabolic process | 3 | 0.0049 | 411.51 | 7.58E-06 | 0.0196 | ALOX15, HPGD |
| GO:2001301 | Lipoxin biosynthetic process | 3 | 0.0049 | 411.51 | 7.58E-06 | 0.0196 | ALOX15, HPGD |
| GO:0050727 | Regulation of inflammatory response | 361 | 0.5848 | 10.259 | 0 | 0.0429 | CCL14, CD200R1, CPB2, |
| GO:0030225 | Macrophage differentiation | 40 | 0.0648 | 46.294 | 0 | 0.0664 | GATA2, LIF, RB1 |

metabolism of angiotensinogen to angiotensins, biosynthesis of DHA-derived SPMs, and biosynthesis of SPMs. Thus, pathways enriched by the DEGs were involved with 2 major processes: the metabolism and synthesis of SPMs and metabolism of the prohormone angiotensinogen. The two genes driving SPM production (ALOX15 and HPGD) were both higher in expression in the healthy animals, suggesting an increase of SPM production in animals that resisted BRD. The two genes driving angiotensinogen metabolism (CPB2 and LOC100139881 (homologous to CMA1 in humans)) were both higher in expression in the healthy animals, suggesting an increase of angiotensinogen metabolism in animals that remained healthy.

## Network analysis of DEGs

Using the DEGs higher in expression in animals that resisted BRD, eighteen proteins were identified by STRING to interact: ABCC4 (LOC528412), ALDH1A1, ALOX15, BMPR1B, BRINP1, CCL14 (LOC100297044), CD200R1 LOC100335828), CHN1, CMA1 (LOC100139881), CPB2, GATA2, GZMB, HPGD, IL5RA, KLF17, LIF, MS4A2, and PRSS45. Functional clustering of these DEGs (labeled red, blue, yellow and green in Fig 4), was performed based upon curated literature and database mining in STRING. This clustering identified distinct functional associations among groups of gene products with regards to 1) (red) proinflammatory molecule inhibition, transcriptional regulation, mast cell induced coagulation, 2) (yellow) leukocyte/lymphocyte proliferation, hematopoietic cell differentiation, inflammatory signaling, 3) (blue) cell cycle regulation, apoptosis, signal transduction, and 4) (green) cellular fatty acid metabolism, prostaglandin-mediated signaling and metabolism (Fig 4).

Using the DEGs higher in expression in the BRD animals (with no more than 5 interactors), protein-protein interactions were analyzed in STRING. Interactions were identified between the DEGs POMC, BGN, MARCO, and MCF2L and two predicted molecules, interleukin (IL)-6 and toll-like receptor 4 (TLR-4) (Fig 5). Specifically, the pro-inflammatory cytokine IL-6 and pro-inflammatory receptor TLR4 were predicted by STRING to be central to the network and to directly interact with 3 DEGs: MARCO, POMC, and BGN.

## RT-qPCR validation of randomly selected DEGs

Relative to healthy cattle, IL5RA, GATA2, ALOX15, and HPGD expression were lower in expression in BRD cattle when validated using qPCR (Fig 6). The genes IL5RA, GATA2, and HPGD were significantly lower in expression in BRD animals (p<0.05) compared to healthy animals. In the case of ALOX15, lower expression in BRD animals relative to healthy cattle was

**Table 2. Significantly enriched biological pathways with associated DEGs in cattle at arrival.**

| Gene Set | Description | Size | P Value | FDR | Genes |
|---|---|---|---|---|---|
| R-HSA-9018896 | Biosynthesis of E-series 18(S)-resolvins | 5 | 1.40E-05 | 0.0209 | ALOX15, HPGD |
| R-HSA-9018679 | Biosynthesis of EPA-derived SPMs | 6 | 2.10E-05 | 0.0209 | ALOX15, HPGD |
| R-HSA-2022377 | Metabolism of Angiotensinogen to Angiotensins | 17 | 0.00019 | 0.0939 | CPB2, LOC100139881 |
| R-HSA-9018677 | Biosynthesis of DHA-derived SPMs | 17 | 0.00019 | 0.0939 | ALOX15, HPGD |
| R-HSA-9018678 | Biosynthesis of specialized proresolving mediators (SPMs) | 19 | 0.00024 | 0.0943 | ALOX15, HPGD |

in the same direction as the results derived from RNA-Seq analysis, but the difference was not statistically significant (p = 0.5623). These expression values were otherwise consistent with our RNA-seq analysis for IL5RA, GATA2, and HPGD.

## Discussion

For several decades research has been conducted to characterize the pathogenesis of BRD, to evaluate immune mechanisms associated with resistance, and to identify methods to prevent the syndrome. In spite of this, BRD continues to be the leading disease in post-weaned beef cattle. Predominantly, research in BRD has emphasized disease pathogenesis characterized by a relatively small number of molecular mechanisms [55] [56] [57]. While these reductionist approaches have been invaluable for recognizing and narrowing important gaps in our understanding of BRD pathogenesis, they fall short in capturing the complexity of gene expression by environment relationships that are no doubt occurring in cattle at risk for BRD. By evaluating the entire blood transcriptome, we simultaneously assessed tens of thousands of gene products and, by extension, a multitude of pathways in order to interrogate the mechanisms that

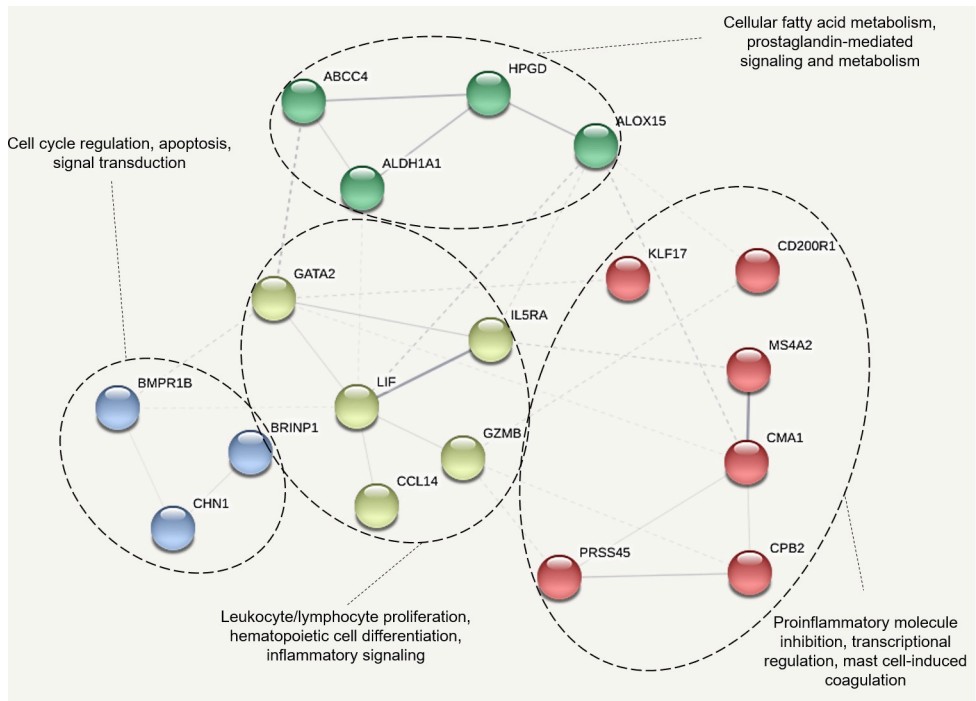

**Fig 4. Protein network of DEGs expressed higher in healthy animals.** K-means clustering performed based on product functionality. Edge (line) thickness between nodes represents the strength of data support for associations.

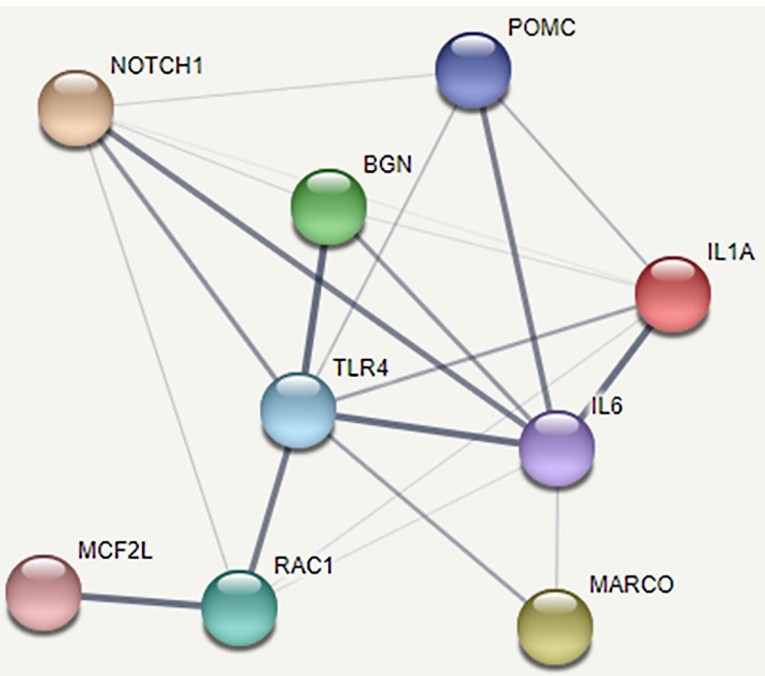

**Fig 5. Protein-protein interaction network with known interactors and DEGs found expressed higher in BRD animals.** Edge (line) thickness between nodes represents the strength of data support for associations. Although TLR-4 and IL-6 were not differentially expressed in this study, the DEGs MARCO, BGN, and POMC possess predicted direct interactions with both molecules.

are operative at arrival in cattle that ultimately develop BRD, and cattle that resist BRD. Recently weaned, transported, and co-mingled cattle are at the highest risk of acquiring BRD in the first three weeks after they are purchased [5] [58]. This report is the first to describe

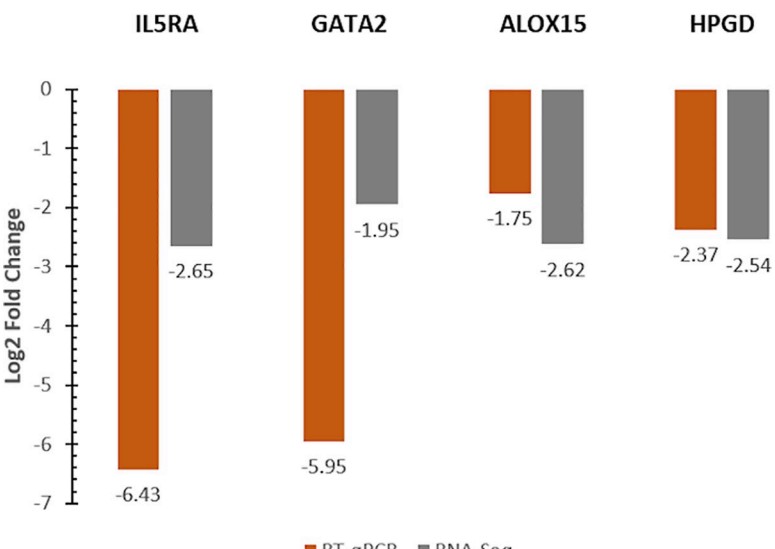

**Fig 6. Gene expression validation via RT-qPCR, compared with relative log2 fold change calculated from RNA-Seq analysis.** RT-qPCR analysis was normalized with the expression levels of the endogenous reference genes DDX31 and UBE2Q1.

whole blood transcriptomes of high-risk beef cattle at arrival, with the aim to identify RNA biomarkers and molecular pathways that contribute to the development of, or resistance to, naturally acquired BRD during this high-risk period.

We are aware that some of the limitations of this study are sample size and the identification of healthy cattle based on treatment metadata. Currently, high-throughput sequencing is a relatively expensive and data-intensive task. However, DEG analysis has been shown to be appropriate with a minimum of 3 biological replicates per experimental group in studies of the effects of sample size on accurate DEG detection [59] [60]. It is reasonable to consider that animals described as healthy (i.e. never showing signs or being treated for clinical BRD) may have been misdiagnosed or possessed mild, self-limiting signs of BRD. Sensitivity of clinical BRD diagnosis remains an ongoing challenge for veterinarians and producers. However, these animals remained viable without treatment and we believe that the inclusion of both treatment history and production (weight gain) outcome of animals placed into each experimental group improved the accuracy of our phenotypic classification [16] [61]. Additionally, previous studies have shown that decreased average daily weight gain is associated with increased morbidity and lung lesions present via thoracic ultrasonography [55] [62] [63].

The objective of this study was to evaluate the whole blood transcriptome of cattle at arrival, and to categorize DEGs and associated pathways enriched at arrival. With this approach our aim was to investigate potential predictive biomarkers measurable at arrival. However, further validation of these gene products and biological pathways as predictive biomarkers in additional cohorts of cattle is warranted. Moreover, while qPCR was performed on a number of DEGs identified in our study, metabolomic and/or proteomic analyses to confirm production of metabolites and proteins predicted by our assessment would have further strengthened this study, and such evaluations should be the focus of future research. Additionally, serial sampling of cattle, starting at arrival, could better define the relationship of gene expression levels and associations with disease status over time.

## Specialized pro-resolvin mediator production genes were expressed higher at arrival in cattle that resist BRD

Specialized pro-resolvin mediators (SPMs) are a class of signaling molecules produced in leukocytes and macrophages, derived from the metabolism of polyunsaturated fatty acids (PUFAs) [64] [65]. During the onset of inflammation, mature leukocytes and macrophages undergo mediator class switching from the production of pro-inflammatory lipid mediators, such as prostaglandins and leukotrienes, to the biosynthesis of SPMs, such as lipoxins and resolvins, to maintain cellular homeostasis and prevent development of chronic inflammation [65] [66] [67]. Two genes involved with SPM production were lower in expression in animals that developed BRD, relative to animals that remained healthy: ALOX15 and HPGD. The gene ALOX15 encodes arachidonate 15-lipoxygenase, an enzyme that is expressed in macrophages, neutrophils, and airway epithelial cells, and has been shown to directly regulate inflammation, innate immunity, and epithelial wound healing [68] [69] [70]. Studies of the immunological response to *Mycobacterium avium* sub. *paratuberculosis* in cattle have demonstrated that infected animals exhibit reduced expression of ALOX15, which is considered relevant to disease persistence and chronicity [71] [72]. The gene HPDG encodes 15-hydroxyprostaglandin dehydrogenase, an enzyme involved in the metabolism of prostaglandins and antagonism of the pro-inflammatory enzyme cyclooxygenase-2 (COX-2) [73] [70] [74] [75].

In our study, ALOX15 and HPGD were significantly higher in expression in cattle that resisted BRD. These genes are both involved in the production of SPMs derived from eicosapentaenoic acid (EPA), specifically the production of E-series 18(S)-resolvins. E-series

resolvins are synthesized during inflammation and infectious processes, to modulate leuko-cyte-driven tissue injury and proinflammatory cytokine production [76] [77]. These resolvins are produced through the modification of EPA by endothelial cells in the presence of aspirin via acetylation of COX2 or through cytochrome p450 conversion, transforming cellular enzy-matic activity by blocking prostaglandin synthesis and allowing for inflammatory resolution [77] [78] [79]. In both murine and human models, resolvin production and enhanced expres-sion of SPM genes has been shown to be protective against acute lung injury and infectious disease, leading to increased patient survivability in the face of acute respiratory disease [80] [81] [82]. Based on our results, the direct actions of ALOX15 and HPGD and downstream pro-duction of resolvins may serve as protective components in beef cattle and play a role in BRD resistance.

Both aspirin and vitamin E metabolites have been experimentally shown to induce SPM gene expression [77] [83]. Interestingly, vitamin E supplementation and adjunct therapeutic use of aspirin have been associated with increased survivability and decreased treatment cost of clinical BRD, although the mechanistic basis of these protective effects have not been vali-dated [84] [85]. Though their adjunct use is not considered a replacement therapy for the metaphylactic use of antimicrobials, the use of aspirin and vitamin E supplementation may enhance SPM production in beef cattle at arrival, while also limiting prostaglandin synthesis. To our knowledge, trait loci and SNP research have not discovered associations between BRD and SPM gene expression.

### Linking angiotensinogen metabolism, mast cell activation, and physiological signaling of inflammation in cattle that resist BRD

Angiotensin II is a circulating peptide hormone classically recognized for its role to modify blood pressure through various mechanisms. Recent discoveries have demonstrated that extra-renal production of renin and localized conversion of angiotensin II may be driven by regional mast cells [86] [87] [88], indicating involvement of the hormone in leukocyte-mediated responses. In this study, cattle resistant to BRD exhibited higher expression of LOC100139881 (CMA1) and CPB2, which encode mast cell protease 1 and carboxypeptidase B2, respectively (Table 2). Mast cell protease 1 and carboxypeptidase B2 are bioactive enzymes necessary for angiotensin II conversion and fibrin regulation via mast cells, during vascular insult and angiogenesis[89] [90] [91]. Previous studies have shown that polymorphisms and abnormal gene expression of CMA1 and MS4A2 are associated with concurrent inflammatory airway disease [92] [93]. Additionally, dysregulation of angiotensinogen metabolism has been associ-ated with metabolic and cardiovascular dysfunction [94] [95]. Collectively, these enzymes have direct interactions with MS4A2 and associations with KLF17 and CD200R1 (Fig 4, red cluster). MS4A2 encodes the membrane spanning beta-subunit of the high-affinity IgE receptor, which is found on mast cells and basophils. This subunit has been shown to be important for mast cell survival, expression of the high affinity IgE receptor, and amplification of receptor- medi-ated signaling events that lead to production of cytokines including interleukin-4 [96] [97]. The proteins encoded by KLF17 and CD200R1, Kruppel like factor 17 and CD200 receptor 1, are involved with regulation of expression of proinflammatory molecules produced by innate immune and effector cells. These gene products are largely involved with mast cell function and appear to have functional associations with LIF, IL5RA, GZMB, and CCL14 (Fig 4, yellow cluster).

The proteins encoded by LIF, IL5RA, GZMB, and CCL14 are historically known to be involved with the proliferation of leukocytes and lymphoid cells, in addition to hematopoietic enhancement and inflammatory signaling. The collective biological activities of these gene

products may represent a biological shift of active immunological processes in animals that resist BRD, specifically involving defense against extracellular antigens. Many of these gene products, especially IL5RA, MS4A2, CPB2, and LIF, are reported in association with Th2-type responses to airway disease [89] [98] [99] [100]. This may be related to the fact that clinical BRD can be driven by extracellular bacterial infection, other extracellular particles, such as LPS or viral structural proteins, and dry, dusty conditions [101], all factors which might stimulate Th2-type responses. While enhancement of genes involved in Th2-type responses and leukocytic recruitment may serve as necessary protective factors against BRD, the converse may also be true, in which Th2-type responses actually moderate protective Th1-mediated inflammatory responses in animals that ultimately resist BRD.

## Persistent inflammatory responses may lead to the development of BRD

Three major gene products that were identified to be higher in expression in BRD cattle are related to the process of pathogen recognition and organism killing: MARCO, POMC, and BGN. The pattern-recognition receptor (PRR) protein encoded by MARCO is a scavenger receptor of macrophages that binds to lipoproteins, Gram-positive, and Gram-negative bacteria [102]. Proopiomelanocortin, encoded by POMC, is a preproprotein that can be converted post-translationally into several bioactive peptides, including α-MSH [103]. α-MSH is an antimicrobial peptide that acts on bacterial and fungal organisms. Biglycan encoded by BGN, is a proteoglycan found in macrophages that stimulates the production of pro-inflammatory cytokines, such as TNF-α and MIP-2, when TLR-2/4 are stimulated by pathogen-associated molecular pattern (PAMP) molecules [104] [105].

Predicted interactions from our analysis indicate that IL-6 and TLR-4 activity is enhanced at-arrival in animals that developed BRD (Fig 5). TLR-4 is the cognate receptor for lipopolysaccharides (LPS) derived from Gram-negative bacteria and viral structural proteins [106]. TLR-4 activation, in turn, induces the expression of IL-6, a pro-inflammatory cytokine responsible for the stimulation of acute phase proteins and leukocyte production [107]. Collectively, these signaling pathways enhance antimicrobial functions in macrophages and dendritic cells [108] [109] [110]. While expression of IL-6 and TLR-4 was not altered in our study, the pattern of expression suggests that animals that acquired BRD were actively combating respiratory disease agents at arrival and did not possess active molecular pathways necessary for mitigating prolonged inflammation. Our main findings suggest that the genes driving the two enriched anti-inflammatory processes of SPM production and angiotensinogen metabolism are underexpressed at arrival in cattle that develop BRD. While both healthy and BRD cattle possessed DEGs that represent functional products involved with the immune system and antimicrobial activity, our results suggest that the ability for cattle to regulate inflammatory processes at arrival appears critical in the resistance of clinical BRD.

## Conclusion

We were able to identify several DEGs between animals that resisted and those that naturally acquired BRD. Two major molecular processes were enriched in animals that resisted BRD: the production of anti-inflammatory lipid pro-resolving mediators (resolvins) and the metabolism of angiotensinogen to angiotensins. We identified functional networks of genes higher in expression in healthy cattle when compared to BRD cattle and describe a biological process involving antimicrobial activity in cattle that developed BRD. Several of the DEGs presented in our study serve a functional role in leukocytes and airway epithelial cells. Our analysis, for the first time, identified an approach to identifying whole blood molecular biomarkers with the potential to predict disease risk in beef cattle in the first 28 days after arrival. As our sample

size for this project was relatively small, further research is necessary to validate these biomarkers, biological processes, and biological pathways in additional populations of beef cattle. Future transcriptomic analyses involving both lung and whole blood from animals at arrival may provide a more complete assessment to further validate the DEGs and pathways present in this study.

## Supporting information

**S1 Table. Descriptive statistics of animals selected for sequencing.**
(XLSX)

**S2 Table. Summary statistics of sample reads post-QC.**
(XLSX)

**S3 Table. DEGs and reference genes selected for RT-qPCR, containing forward and reverse primers and product length.**
(XLSX)

**S4 Table. Complete list of DEGs and associated testing variables found in edgeR and DESeq2 analysis.**
(XLSX)

## Acknowledgments

The authors would like to thank the students and staff of the Mississippi Agricultural and Forestry Experiment Station (MAFES) and Mississippi State University Animal and Diary Science Department for their assistance in animal care and sample collection. We would also like to thank Merrilee Thoresen, Daniele Doyle, and Kathleen Barton for their technical assistance and insight throughout this experiment.

## Author Contributions

**Conceptualization:** Matthew A. Scott, Amelia R. Woolums, Cyprianna E. Swiderski, David R. Smith, Brandi B. Karisch.

**Data curation:** Matthew A. Scott, Amelia R. Woolums, Cyprianna E. Swiderski, David R. Smith, William B. Epperson.

**Formal analysis:** Matthew A. Scott, Cyprianna E. Swiderski, Andy D. Perkins, Bindu Nanduri, David R. Smith.

**Investigation:** Matthew A. Scott, Amelia R. Woolums, Bindu Nanduri, David R. Smith, Brandi B. Karisch, William B. Epperson, John R. Blanton Jr.

**Methodology:** Matthew A. Scott, Amelia R. Woolums, Cyprianna E. Swiderski, Andy D. Perkins, Bindu Nanduri.

**Resources:** Amelia R. Woolums, Cyprianna E. Swiderski, Andy D. Perkins, Brandi B. Karisch, William B. Epperson, John R. Blanton Jr.

**Software:** Matthew A. Scott, Cyprianna E. Swiderski, Andy D. Perkins.

**Supervision:** Amelia R. Woolums, David R. Smith, Brandi B. Karisch, William B. Epperson, John R. Blanton Jr.

**Validation:** Matthew A. Scott, Cyprianna E. Swiderski, Andy D. Perkins.

**Visualization:** Matthew A. Scott, Cyprianna E. Swiderski.

**Writing – original draft:** Matthew A. Scott.

**Writing – review & editing:** Matthew A. Scott, Amelia R. Woolums, Cyprianna E. Swiderski, Andy D. Perkins, Bindu Nanduri, David R. Smith, Brandi B. Karisch, William B. Epperson, John R. Blanton Jr.

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
