## [Decision Letter · Decision Letter 0]

29 Oct 2019

PONE-D-19-24105

Whole blood transcriptomic analysis of beef cattle at arrival identifies potential predictive molecules and mechanisms that indicate animals that naturally resist bovine respiratory disease

PLOS ONE

Dear Dr. Scott,

Thank you for submitting your manuscript to PLOS ONE. After careful consideration, we feel that it has merit but does not fully meet PLOS ONE’s publication criteria as it currently stands. Therefore, we invite you to submit a revised version of the manuscript that addresses the points raised during the review process.

We would appreciate receiving your revised manuscript by Dec 13 2019 11:59PM. To enhance the reproducibility of your results, we recommend that if applicable you deposit your laboratory protocols in protocols.io, where a protocol can be assigned its own identifier (DOI) such that it can be cited independently in the future. For instructions see: http://journals.plos.org/plosone/s/submission-guidelines#loc-laboratory-protocols

We look forward to receiving your revised manuscript.

Kind regards,

Juan J Loor

Academic Editor

PLOS ONE

Journal Requirements:

Reviewers' comments:

Reviewer's Responses to Questions

**Comments to the Author**

1. Is the manuscript technically sound, and do the data support the conclusions?

Reviewer #1: Yes

Reviewer #2: Yes

2. Has the statistical analysis been performed appropriately and rigorously? 

Reviewer #1: Yes

Reviewer #2: Yes

3. Have the authors made all data underlying the findings in their manuscript fully available?

Reviewer #1: Yes

Reviewer #2: Yes

4. Is the manuscript presented in an intelligible fashion and written in standard English?

Reviewer #1: Yes

Reviewer #2: Yes

5. Review Comments to the Author

Reviewer #1: Review on manuscript by Scott et al.

Whole blood transcriptomic analysis of beef cattle …

In this manuscript, the authors reported their study to examine and identify differentiated expressed genes in cattle which may play a role in resistance to BRD. They collected blood samples from cattle on arrival, subsequently, they used a set of criteria to identify those animals that are healthy or had BRD, and performed whole transcriptome analysis of their blood samples. The data was then extensively analysed by several platforms to identify differentially expressed genes in the healthy animals which may contribute to their resistance to BRD.

This is a preliminary study to identify potential host factors/immune pathways that could be involved in resistance to BRD. The study was well planned, though sample size used in the study was small as recognized by the authors. The technical part of the transcriptome analysis and date analysis was extensively and well carried out. However, the authors tried to make too much of the findings regarding the role of the DFGs. In this first study, caution should be taken with such small sample size. I have the following comments:

1. It has been well established that bacterial pathogens (and viral pathogens to a minor degree) are the cause of BRD. Even though this study is aimed at the host factors, the authors made no mention of the bacterial pathogens. E.g. What pathogens were recovered from the the animals at necropsy?

2. Along the same line, I suggest the author should consider examining the lung tissue/fluid from animals, both healthy and diseased from BRD. Total transcriptome analysis of DEGs from this location should provide more evidence to host genes and proteins involved in resistance to BRD.

3. They identified five pathways which were over expressed in healthy animals, are there any genes/pathways that are under expressed in the BRD animals? They should look into the lack of an immune response which may contribute to colonization and proliferation of bacterial pathogens in BRD.

4. For a more complete study, additional blood samples should be taken, perhaps weekly, to provide a trend in gene expression, currently, the analysis is based on one data point per animal.

Reviewer #2: The manuscript “Whole blood transcriptomic analysis of beef cattle at arrival identifies potential predictive molecules and mechanisms that indicate animals that naturally resist bovine respiratory disease”

The authors reported the results of RNAseq data regarding the transcriptome of beef cattle that develop BRD compared to beef cattle that don’t develop the disease. In general, the manuscript is well written and rise some interesting novelties to a a better understanding of the BRD.

Some of my comment can help to improve the manuscript.

In the abstract are missing information about the experimental plan. Please add a brief description of your study.

Line 116-137: I find this part difficult to follow, try to simplify that.

Line 134. How did you make the samples selection?

Line 260-263: it is obvious that using the 36 DEG you have success in the clustering analysis.

Line 306, 322, 336, 377 …. the use of increased or decreased in the contest of your study is not correct. Instead, you have a higher/lower expression of the gene. Please correct throughout the manuscript.

Line 363-375: in this paragraph you correctly described some limitation of your study mainly about sample size. For my experience the sample number of your study is fine. The real limitation of your study is that you only analyze transcriptome. You find some interesting metabolic pathway but without metabolic confirmation. Please add a clear statement about this aspect as limitation of your study.

Line 418-444: this part must be reduced. Consider also that it is out of contest because you are working with Whole blood cells not with liver!

Line 459: what do you mean with “unchecked”?

CONCLUSION: specified more clearly that your study is related to circulating leucocytes

Figure 1. change disease to BRD

Figure5 insert in the figure or in the caption that il6 and TLR4 are only predicted in your study.

6. PLOS authors have the option to publish the peer review history of their article (what does this mean?). If published, this will include your full peer review and any attached files.

Reviewer #1: No

Reviewer #2: No

---

## [Author Response · Author response to Decision Letter 0]

5 Dec 2019

All responses made may be found in the attached file labeled "Response to Reviewers." No changes were made to our cover letter or financial statement.

---

## [Decision Letter · Decision Letter 1]

20 Dec 2019

Whole blood transcriptomic analysis of beef cattle at arrival identifies potential predictive molecules and mechanisms that indicate animals that naturally resist bovine respiratory disease

PONE-D-19-24105R1

Dear Dr. Scott,

We are pleased to inform you that your manuscript has been judged scientifically suitable for publication and will be formally accepted for publication once it complies with all outstanding technical requirements.

With kind regards,

Juan J Loor

Academic Editor

PLOS ONE

Additional Editor Comments (optional):

Reviewers' comments:

Reviewer's Responses to Questions

**Comments to the Author**

1. If the authors have adequately addressed your comments raised in a previous round of review and you feel that this manuscript is now acceptable for publication, you may indicate that here to bypass the “Comments to the Author” section, enter your conflict of interest statement in the “Confidential to Editor” section, and submit your "Accept" recommendation.

Reviewer #1: All comments have been addressed

Reviewer #2: All comments have been addressed

2. Is the manuscript technically sound, and do the data support the conclusions?

Reviewer #1: Yes

Reviewer #2: Yes

3. Has the statistical analysis been performed appropriately and rigorously? 

Reviewer #1: Yes

Reviewer #2: Yes

4. Have the authors made all data underlying the findings in their manuscript fully available?

Reviewer #1: Yes

Reviewer #2: Yes

5. Is the manuscript presented in an intelligible fashion and written in standard English?

Reviewer #1: Yes

Reviewer #2: Yes

6. Review Comments to the Author

Reviewer #1: I am satisfied with the revision, comments and suggestions were addressed satisfactorily. This study should lead into further investigations.

Reviewer #2: the authors responded satisfactorily to all my requests

no further comments

7. PLOS authors have the option to publish the peer review history of their article (what does this mean?). If published, this will include your full peer review and any attached files.

Reviewer #1: No

Reviewer #2: No

---

## [Editor Report · Acceptance letter]

27 Dec 2019

PONE-D-19-24105R1 

Whole blood transcriptomic analysis of beef cattle at arrival identifies potential predictive molecules and mechanisms that indicate animals that naturally resist bovine respiratory disease 

Dear Dr. Scott:

I am pleased to inform you that your manuscript has been deemed suitable for publication in PLOS ONE. Congratulations! Your manuscript is now with our production department. 

With kind regards,

on behalf of

Dr. Juan J Loor 

Academic Editor

PLOS ONE